# Active and Passive Immunization with an Anti-Methamphetamine Vaccine Attenuates the Behavioral and Cardiovascular Effects of Methamphetamine

**DOI:** 10.3390/vaccines10091508

**Published:** 2022-09-09

**Authors:** Colin N. Haile, Kurt J. Varner, Xia Huijing, Reetakshi Arora, Frank M. Orson, Thomas R. Kosten, Therese A. Kosten

**Affiliations:** 1Department of Psychology/TIMES, University of Houston, Houston, TX 77204, USA; 2Department of Pharmacology and Experimental Therapeutics and the Cardiovascular Center of Excellence, LSUHSC, New Orleans, LA 70112, USA; 3The Michael E DeBakey Veteran’s Affairs Medical Center, Houston, TX 77030, USA; 4Menninger Department of Psychiatry and Behavioral Sciences, Baylor College of Medicine, Houston, TX 77030, USA; 5Immunology Allergy & Rheumatology, Department of Medicine, Baylor College of Medicine, Houston, TX 77030, USA

**Keywords:** conjugate vaccine, entolimod, methamphetamine use disorder, locomotor activity, drug reinstatement, medication development

## Abstract

Background: Methamphetamine use disorder (MUD) is a growing health concern with no FDA-approved treatment. The present series of studies build upon our previous work developing an anti-methamphetamine (MA) vaccine for MUD. We determined the effects of a formulation that included tetanus-toxoid (TT) conjugated to succinyl-methamphetamine (TT-SMA) adsorbed onto aluminum hydroxide (alum) in combination with the novel Toll-Like Receptor-5 agonist, entolimod. Methods: Mice were vaccinated (0, 3, 6 weeks) with TT-SMA+alum and various doses of entolimod to determine an optimal dose for enhancing immunogenicity against MA. Functional effects were then assessed using MA-induced locomotor activation in mice. Experiments using passive immunization of antibodies generated by the vaccine tested its ability to attenuate MA-induced cardiovascular effects and alter the reinforcing effects of MA in an MA-induced reinstatement of a drug seeking model of relapse in male and female rats. Results: Antibody levels peaked at 10 weeks following vaccination with TT-SMA+alum combined with entolimod (1, 3 and 10 µg). MA-induced locomotor activation was significantly attenuated in vaccinated vs. unvaccinated mice and antibody levels significantly correlated with ambulation levels. Passive immunization decreased mean arterial pressure following MA dosing in rats of both sexes but did not alter heart rate. Passive immunization also attenuated the ability of MA to reinstate extinguished drug-seeking behavior in male and female rats. Results support further development of this vaccine for relapse prevention for individuals with MUD.

## 1. Introduction

The 2020 National Survey on Drug Use and Health estimates that 2.5 million individuals used methamphetamine (MA) in the past year, and overdose deaths increased by 180% from 2015 to 2019 [1,2]. The urgent need for efficacious treatments for MA use disorder (MUD) is further supported by the nearly 5-fold increase in urine positivity rates for MA among those entering treatment from 2013 to 2019 [3,4,5]. There are no FDA-approved pharmacotherapies for MUD and clinical trials assessing potential treatments suffer from non-compliance, high drop-out rates, and substantial variation in pharmacotherapy responses suggesting underlying differences in MUD neurobiology and the need to match pharmacotherapies to specific MUD subgroups [6,7]. Immunotherapies for MUD can potentially surmount these compliance hurdles because they are long-lasting. Vaccines also render the individual variations in MA neurobiology and need for matching medication to MUD subgroups unimportant because the MA is prevented from entering the brain. Vaccines developed for MUD should produce antibodies (AB) that target MA and sequester it in the peripheral circulation to prevent its penetration into the brain. Delay and reduction of MA entering the brain can diminish MA reinforcing effects that, in turn, reduce MA use and help prevent relapse after a period of abstinence. 

Anti-MA vaccines that vary in hapten design, carrier proteins, linkers between the hapten and carrier protein, as well as adjuvants, have been shown by several laboratories to alter the behavioral and functional effects of MA [8,9,10,11,12,13,14,15,16,17,18,19,20,21,22,23]. A monoclonal AB for overdose reversal also blocks MA-induced behavioral effects in animals [24]. MA vaccines using keyhole limpet hemocyanin (KLH) as the carrier protein generate substantial AB titers with good affinity for MA [11] and attenuate behavioral effects in mice [14]. Because KLH is a large and structurally complex protein, conjugate vaccines using KLH are difficult to manufacture. Thus, we tested a vaccine in which the tetanus toxoid (TT) carrier protein was conjugated to a succinyl-MA (SMA) hapten (TT-SMA). Tests in mice show that the TT-SMA vaccine generated substantial AB levels with good affinity and lead to decreased brain MA levels after acute administration as well as to attenuated behavioral effects including acquisition and reinstatement of MA place conditioning [17,21]. However, we learned from our previous human studies of an anti-cocaine vaccine that using alum as the only adjuvant is insufficient to generate sufficient AB titers for at least one-third of individuals with cocaine use disorder [25,26]. We therefore conducted studies to assess whether other adjuvants enhance immunogenicity of TT-SMA. 

Adjuvants targeting Toll-like receptors (TLRs) are considered a potent way to enhance innate and adaptive immune responses to antigens [27,28,29] and to boost the immune response [30]. A variety of TLR agonists have been tested: TLR-9 (CpG ODN 1826), TLR- 4, and TLR-2 [19,29,31]. The TLR-9 adjuvant is DNA based, while the TLR-2 and TLR-4 adjuvants are based on various lipids [32,33,34,35]. The TLR-5 adjuvant, entolimod, is derived from the protein flagellin on Salmonella bacteria and it has excellent safety at doses up to 30 µg in humans [36,37]. The present study assessed the efficacy of entolimod at much lower doses of 0.03 to 10 µg in combination with TT-SMA to produce anti-MA AB titers and attenuate MA-induced locomotor activity in mice. We further tested if passive immunization with anti-MA AB would reduce the cardiovascular effects of MA as well as its ability to reinstate extinguished lever press responding for MA in rats, an animal model of relapse [38,39,40]. 

## 2. Materials and Methods

### 2.1. Animals, Housing, and Drug

Fifty-eight female BALB/c mice obtained from Charles River Laboratories (Wilmington, MA, USA) were group housed (5 per cage) under temperature- and humidity-controlled conditions at the Michael E. DeBakey VAMC vivarium at Baylor College of Medicine with ad libitum access to food (Harlan Teklad, Envigo, Indianapolis, IN, USA) and water. Mice weighed 20–25 grams at the start of the experiments. Thirty-five mice were used to measure antibody levels and 23 were used for the locomotor test. Eighty female Fischer 344 rats (Charles River, 250–300 g) used to generate anti-MA antibodies for passive immunization were group-housed (5 per cage) under temperature- and humidity-controlled conditions at the Michael E. DeBakey VAMC vivarium. Ten male and 11 female Sprague-Dawley rats (Charles River) were housed singly under temperature- and humidity-controlled conditions at the Louisiana State University Health Sciences Center (LSUHSC) animal facility with ad libitum access to food and water. Male rats weighed 170–230 g and female rats weighed 160–190 g at the start of the experiment and were used for the cardiovascular study. Four Sprague-Dawley rats (2 males; 2 females; Charles River) were housed singly under temperature- and humidity-controlled conditions at the University of Houston animal facility with ad libitum access to food and water except during food training when they were restricted to about 90% of free-feeding body weight. Male rats weighed 250–300 g and female rats weighed 200–250 g at the start of the experiment and were used for the self-administration study. All experimental procedures adhered to the guidelines defined in the NIH Guide for the Care and Use of Laboratory Animals and were approved by the IACUCs at Baylor College of Medicine, LSU Health Sciences Center, and the University of Houston. Methamphetamine hydrochloride (MA: Research Triangle Institute, Research Triangle, NC, USA and Sigma-Aldrich, St. Louis, MO, USA) was dissolved in sterile saline. Drug (2 mg/kg, IP) was administered at a volume of 10 mL/kg for the mouse locomotor study. MA was administered IV at a dose of 2 mg/kg/injection for the rat cardiovascular study and in a volume of 1 mL/kg. For the rat self-administration reinstatement tests MA was also administered in a volume of 1 mL/kg. 

### 2.2. Vaccine Preparation and Administration

As previously detailed [17,21], the hapten, succinyl MA (SMA) (Research Triangle Institute, Research Triangle Park, NC, USA) was conjugated to TT in PBS (TT-SMA) (MassBiologics, Boston, MA, USA). After adjusting the pH to 7.5 the final concentration of the TT conjugate was 9.8 mg/mL. Dose optimization of the entolimod was performed in female BALB/c mice with 32 µg of TT-SMA in combination with varying doses of entolimod in the presence and absence of alum (1500 µg) (Brenntag, Inc., Reading, PA, USA). The final vaccine formulation consisted of TT-SMA and alum with the optimal dose of entolimod (1 µg; see Results). The initial intramuscular (IM) vaccination (week 0) was followed by two booster IM vaccinations at week 3 and 6. Since the optimal entolimod dose was 1 µg, all subsequent studies used 1 µg entolimod formulated with 32 µg of TT-SMA and 1500 µg of alum. Blood (~100 µL) was collected by saphenous vein puncture from all mice at the specified time points and allowed to clot for 2 h then centrifuged (4000 rpm for 15 min) to collect the sera. Serum samples were stored at 4 °C until analyzed for anti-MA AB. 

### 2.3. Quantification of Anti-Methamphetamine Antibody Levels

Enzyme linked immunosorbent assays (ELISAs) were used to measure anti-MA specific AB concentrations. The ELISA plates (Immulon 2HB, Daigger, Vernon Hills, IL, USA) were coated overnight in carbonate buffer (0.05 M; pH 9.6) using fish gelatin, a heterologous carrier protein, as the conjugate partner for SMA. The plates were washed and then blocked for 2 h at room temperature using PBS-Tween. Pooled or individual serum samples were added to plates in 2-fold serial dilutions starting at 1K/3K in PBS-Tween (0.1%) and incubated for 2 h. Plates were washed with PBS-Tween prior to adding goat anti-mouse IgG conjugated to HRP (Southern Biotech, Birmingham, AL, USA). Plates were incubated for another 60 min and washed before adding substrate (tetramethylbenzidine, Sigma-Aldrich, St. Louis, MO, USA). Plates were incubated for 45 min in the dark prior to stopping the reaction with 1M HCl. The plates were read with a microplate reader (iMark Microplate Absorbance Reader) using Microplate Manager v6.1 software (Bio-Rad, Hercules, CA, USA) to assess the optical density (OD) of each cell on the plate. The background AB binding (ranging <0.1 at lowest dilution) to the carrier alone was subtracted from each sample to ensure the quantification of MA specific AB. 

### 2.4. Methamphetamine-Induced Locomotor Activity in Mice

Locomotor activity of non-vaccinated mice (N = 10) was compared to that of actively immunized mice (N = 8) following administration of MA (2 mg/kg, IP) as well as to a group of non-vaccinated mice administered vehicle (0 mg/kg; N = 5). Mice were first habituated to the locomotor apparatus (Opto-M3; Columbus Instruments; Columbus, OH, USA) and to the injection procedure (saline) 4–5 times prior to MA drug challenge so that baseline activity did not differ between groups. Locomotor activity was assessed by placing the mice into individual acrylic test chambers (27.5 in. × 27.5 in. × 8.0 in). Infrared sensors located 1″ above the floor spaced at 2.25″ intervals around the perimeter of the open-field test apparatus were used to tabulate ambulatory counts of mouse movements using a software program (Columbus Instruments) installed on a PC computer. Baseline activity was recorded for 50 min and then for another 90 min after the MA injection in 10-min intervals. 

### 2.5. Passive Immunization

Rats used to generate anti-MA antibodies for the cardiovascular study were administered 100 µg TT-SMA (IP) + CFA (Complete Freund’s Adjuvant, 60 µg, SC) followed by a boost with IFA (Incomplete Freund’s Adjuvant, 40 µg, SC) two weeks post initial vaccination. Rats used to generate anti-MA antibodies for the self-administration study were immunized as described above under vaccine preparation and administration. Trunk blood was collected from each rat at ten weeks and processed to obtain serum. Serum anti-MA antibody levels were quantified by ELISA and confirmed prior to administration.

### 2.6. Intravenous Cannula and Telemetry Probe Implantation

Rats were anesthetized with a mixture of ketamine–xylazine (90/10 mg/kg, IP). The surgical areas in the left groin and dorsal nape of the neck were shaved and sterilized with Betadine and a sterile drape placed over the rat. The rat was placed on a heating pad to maintain body temperature at 38 + 1 °C. The depth of anesthesia was monitored by lack of movement in response to foot and tail pinch or lack of corneal reflex. On Day-7, the left femoral vein and artery were isolated through an incision in the left groin. A saline (0.9% sterile) filled cannula (PE 50, Intramedic, Clay Adams, Thomas Scientific, Swedesboro, NJ, USA) was inserted (1–2 cm) into the femoral vein through an incision cut into the vein. The cannula was then secured by tying the cannula to the vessel distal to and proximal to the insertion site. The free end of the cannula was tunneled subcutaneously to the nape of the neck, exteriorized, and sutured in place with 5-0 silk sutures. After verifying cannula patency by withdrawing blood, the cannula was filled with sterile saline containing sodium heparin (200 U/mL) and capped with a stainless-steel pin. The IV cannula was flushed with 0.1–0.2 mL of heparinized saline on day-4 and day-2. The neck wound was then closed with silk sutures. The patency of the cannula was verified 2 days before and on the day of the experiment (Day 0) by observing the pressor response and bradycardia elicited by a bolus IV injection of phenylephrine (2–6 μg).

The cannula of a sterile radio telemetry probe (HSD-10, Data Sciences International, St. Paul, MN, USA) was introduced into the femoral artery of the left leg through a small incision in the artery. After advancing the cannula ~2 cm, the cannula was secured with tissue glue at the insertion site and sutured around the cannula and the artery distal to and proximal to the insertion point. The body of the telemetry probe was then placed in a pocket under the skin of the left flank just anterior to the hip. The wound in the groin was closed with 5-0 silk sutures. Rats were allowed a week to recover and external sutures were removed after 7 days. 

### 2.7. Tests Following Methamphetamine Administration

On Day-2 (for testing of cannula patency) and on the days of the experiment, the telemetry probes were activated by passing a magnet near the animal’s flank. Arterial pressure (systolic, diastolic, and mean) was recorded (1000 Hz) by the probes from the conscious freely moving animals by receivers placed under the home cage and stored on a PC. Heart rate was derived automatically from the arterial pulse. After achieving stable arterial pressure and heart rate recordings, 2–6 μg of phenylephrine (Sigma-Aldrich) in 2–6 μL saline was loaded into the distal end of the injection cannula using a 25 μL Hamilton micro-syringe. The phenylephrine was infused over 2 to 3 s with 0.1 mL of saline and the arterial pressure and heart rate responses recorded. On days when only cannula patency was tested the injector line was disconnected and the cannula closed. On experimental days, the injector line remained attached and the rat was monitored for at least 1-h to obtain stable baselines. Once a stable baseline was achieved, the rats were given a 2 mg/kg IV dose of MA (in 25–35 μL of saline). The MA was loaded into the distal end of the injection line and flushed with 0.1 mL of saline. The MA dosing was repeated 1 h and 2 h after the first dose. Once the blood pressure and heart rate returned to within 10% of baseline after the third dose, the IV injection line was disconnected and the catheter sealed. After obtaining pre-antibody MA measures on arterial pressure and heart rate, each rat received an IP injection (3 mL) of the serum containing MA antibodies. The injection was infused over 1 to 2 min. The rats were then placed in their home cages and monitored for the next 2 to 3 h. Twenty-four hours later, the rats were given a second 3 mL injection of serum. The rats were again monitored for the next 2 to 3 h to assess if any adverse effects of the serum injection occurred. The following day the rats’ home cages were placed back on the telemetry plates and the IV injector lines reconnected. Baseline values of arterial pressure and heart rate were recorded until stable baselines were achieved (approximately 1 h). Three MA injections (2 mg/kg, IV) were again given 1 h apart as described above and measures recorded. 

### 2.8. Intravenous Catheter Implantation and Methamphetamine Self-Administration Training

This study assessed the effects of passive immunization with anti-MA AB sera on reinstatement of drug-seeking behavior in rats trained to self-administer IV MA. Prior to surgery, rats were trained to lever press for food (30 min sessions) under a continuous reinforcement schedule until consistent lever press responding was achieved (e.g., obtaining the maximum number of 50 food reinforcers on two occasions in 30 min or less). Rats were then surgically implanted with indwelling intravenous jugular catheters as we conducted previously with slight modifications [41]. The rat was anesthetized with isoflurane anesthesia (2–3%) and surgical sites (right ventral neck area and dorsal midline mid-scapular) prepared by first shaving then cleaning the areas with Betadine solution followed with 70% ethanol. The rat was placed supine and the jugular vein located and catheter inserted into the right vein and secured with 2.0 silk. A dorsal mid-scapular subcutaneous space was then dissected approximately 2.5 cm in diameter for ventral access button placement. Using a curved blunt hemostat, a subcutaneous passage was made for the catheter from the jugular incision to the mid-scapular incision. The catheter was then pulled through the mid-scapular incision site to the jugular neck area and clamped with a needle nose hemostat. A vascular access “button” was then attached to the end of the catheter to permit use of a miniature PinPort system (Instech Laboratories, Inc., Plymouth Meeting, PA, USA). The incisions were then sutured closed using monofilament polypropylene suture. Rats were allowed at least 7 days recovery before beginning MA self-administration sessions. Catheters were flushed daily with a heparinized saline solution. 

MA self-administration sessions were conducted using standard operant chambers (Coulbourn Instruments, Holliston, MA, USA) enclosed in sound-attenuating cubicles (Coulbourn Habitest isolation cubicle). Each chamber was equipped with two levers located on either side of an access area into which food pellets could be delivered. Within the operant chamber were a house light and two sets of three, colored cue lights, one above each lever. Graphic State Notation (version 4.0; Coulbourn Instruments) was installed on a PC and used to program experimental and stimulus parameters and tabulate data.

MA self-administration training sessions (2 h) were initiated after recovery from surgery. Rats were placed in the operant chambers and their catheter/cannula system was attached to a syringe pump system that consisted of an infusion pump (Razel model A) with a 20-mL glass syringe connected to a tether (Instech Laboratories, Inc., Plymouth Meeting, PA, USA). The tether was connected to the animals’ PinPort assembly using Tygon tubing protected by a metal spring and secured to the animal’s back port via a Luer-lock attachment. Initially, rats were given one priming infusion and then one depression of the active lever resulted in an intravenous infusion of 100-µL injection of MA under a fixed ratio 1 (FR1) schedule of reinforcement in which a dose of 0.125 mg/kg was delivered per infusion. The MA infusion was delivered over a 10 s period, followed by a 20 s time-out. Cue lights were illuminated above the active and inactive levers. Cue and house lights were turned off during the entire infusion and time-out periods. Active lever presses emitted during the infusion time or during the time-out period were tabulated but did not result in any further MA delivery. Presses on the inactive lever were tabulated but had no programmed consequences. When MA showed control over behavior under the FR1 schedule of reinforcement, the FR requirement was raised to FR2, the schedule used for the remainder of the experiment. Once consistent self-administration responding was shown (number of reinforcers earned showed <20% variability over two consecutive days), tests for dose-related responding to MA were initiated followed by reinstatement tests as described below.

### 2.9. Reinstatement Tests

Reinstatement tests were performed similar to our previous study [42]. Extinction sessions in which saline replaced MA infusions were conducted until active lever presses were <10% maintenance response levels. Following extinction, reinstatement tests were initiated by administering an injection of MA (0.125 mg/kg; IP) and active and inactive lever presses recorded over 2 h. Following reinstatement tests, successive extinction sessions continued until lever presses were again at <10% maintenance response levels. Rats were then passively immunized with anti-MA AB at an average of 105.12 µg/gm, one injection per day over two days. The rats then underwent a second set of reinstatement tests. Active lever press data from the final two extinction tests (Ext1 and Ext2) conducted prior to the two reinstatement tests (R1 and R2; pre- and post-immunization, respectively) were compared to test the ability of passive immunization to block MA-induced reinstatement of responding. 

### 2.10. Data Analysis

ELISA data were analyzed using SigmaPlot (Systat Software Inc., Chicago, IL, USA) with subtraction of background AB binding to the carrier alone for each sample. Locomotor activity was analyzed using a repeated measures ANOVA with group (vaccine vs. no vaccine) and time as the repeated factor. Regression analysis was used to determine potential associations between total ambulatory counts and serum antibody levels. Cardiovascular measures (change from baseline to peak) were analyzed using Statistica (v. 13.2.92.5) with Vaccine Treatment (before vs. after) and dosings as the factors in the ANOVA. Separate analyses were conducted by sex due to the large baseline differences in cardiovascular measures and body weight. A t-test for correlated samples was used for the reinstatement tests in which the numbers of active lever presses emitted before immunization were compared to the numbers emitted after immunization.

## 3. Results

### 3.1. Anti-MA Response to Different Entolimod Doses

The entolimod doses that produced maximum peak AB concentrations against MA were determined using seven groups (n = 5/group) of BALB/c female mice: seven groups received 32 µg of TT-SMA and 1500 µg alum plus one of the following entolimod doses 0.01, 0.03, 0.1, 0.3, 1, 3, and 10 µg as in Figure 1A,B which shows the peak anti-MA AB concentrations at week 10 were quite similar for 1, 3, and 10 µg of entolimod. The 1 µg entolimod dose was selected for subsequent experiments because, by week 10, the 1 µg dose led to peak AB levels equivalent to the two higher doses. 

### 3.2. MA Induced Locomotor Activity

The locomotor test was conducted in mice at 7 weeks after the initial immunization. Ambulatory locomotor counts are low during the 50 min pre-drug habituation session as shown in Figure 2A (-5-0 10 min blocks). Activity (1-9 10 min blocks) levels initially increase dramatically in the groups administered MA (2 mg/kg) and then decrease across the session as supported by the significant Time effect, F(9486) = 27.15; *p* < 0.0001. Activity levels are lower in the vaccinated mice compared to the non-vaccinated mice administered MA. This is supported by the significant Vaccine effect, F(2,54) = 15.01; *p* < 0.0001. The significant Vaccine X Time interaction likely reflects that activity levels of vaccinated mice administered MA begin to return to baseline more quickly than that of the non-vaccinated mice administered MA, as well as to the lack of change in the low activity levels across the session in the vehicle-injected group, F(18,486) = 4.90; *p* < 0.0001. In vaccinated mice, lower total locomotor activity induced by MA significantly associates with higher AB levels and accounts for 62% of the variance in locomotion (*p* < 0.05) as shown in Figure 2B.

### 3.3. Passive Immunization and MA Cardiovascular Effects

Changes in mean arterial pressure (MAP) and heart rate (HR) elicited by three consecutive dosings of MA (2 mg/kg; IV) were assessed in male and female rats prior to and after passive immunization with MA antibodies. The MA dosings were separated by 1 h and the post-immunization MA challenge began 48 h after initiating the immunization procedure. The passive immunization resulted in a mean of 463 µg/mL (SD = 31.5) of anti-MA AB concentration within the rodents, which was a mean percentage transfer of 28% (SD = 0.02) compared to the infusion AB concentration. Figure 3A,B shows the change from baseline levels in MAP elicited by MA in male and female rats, respectively. Pressor responses were significantly reduced across the three MA dosings in male rats, F(2,40) = 25.34; *p* < 0.0001, and in female rats, F(2,36) = 27.41; *p* < 0.0001. After immunization, the pressor responses to MA were significantly smaller than before immunization as supported by the significant Vaccine effects in both male rats, F(1,20) = 12.57; *p* < 0.005, and female rats, F(1,18) = 6.04; *p* < 0.05. Figure 3C,D shows the change from baseline levels in HR elicited by MA in male and female rats, respectively. The increases in HR levels were significantly reduced across the three MA dosings in both male rats, F(2,40) = 6.09; *p* < 0.005, and female rats, F(2,36) = 3.81; *p* < 0.05. There were no significant effects of passive immunization on HR levels in male or female rats (*p*’s > 0.10). There were also no significant interactions of Dosing by Vaccine treatment for either measure in either sex (*p*’s > 0.10). 

### 3.4. Passive Immunization and Drug-Induced Reinstatement

Passive immunization with anti-MA AB blocked drug-induced reinstatement of MA-seeking behavior. Figure 4 shows the active lever press responses from the four rats (two females: M4 and M11; and two males: M17 and M22) that were tested at baseline and after passive immunization with antibodies generated from rats administered the TT-SMA vaccine. After rats underwent extinction (EX1-EX2), they received an IP injection of MA (2 mg/kg, R1) that produced an increase in lever pressing (i.e., drug-seeking). Following this initial reinstatement test (R1), extinction training resumed and the rats were passively immunized with anti-MA AB as described above. Rats then received another dose of MA (R2). As seen in Figure 4, passive immunization with anti-MA significantly blocked drug-seeking behavior, t(3) = 2.95; *p* < 0.05. All rats emitted fewer active lever presses after immunization (R2) compared to before treatment (R1) with three out of four rats emitting no lever presses. The one rat that did lever press after immunization (M4) emitted the same number of responses seen during the first extinction (EX1) session prior to treatment and was the only rat that responded during the second extinction (EX2) trial after treatment. Rat M4 also had the lowest anti-MA AB levels of the four rats, which had anti-MA AB blood levels of 75, 110, 200, and 210 ng/mL (M4, M11, M17, and M22, respectively). 

## 4. Discussion

The results of the present study demonstrate that the entolimod adjuvant enhances the ability of the anti-MA vaccine to produce MA-specific ABs and attenuates the behavioral effects of MA in mice and rats. The vaccine used in the present study utilizes the tetanus-toxoid (TT) protein carrier conjugated to the succinyl-methamphetamine (SMA) hapten. Previously, we found that TT-SMA combined with a different adjuvant (E6020) produced MA-specific antibodies at threefold greater peak titer levels compared to TT- SMA without E6020 [21]. Using equilibrium dialysis to test binding across a wide drug concentration range, we also showed that the average affinity of TT-SMA is 82 nM (range: 20–110 nM). In that study, we also demonstrated that the vaccine block MA-induced behavioral effects in mice like our prior vaccine constructed with KLH as the carrier protein [14]. We have now expanded upon this research to show that active immunization with TT-SMA plus entolimod attenuates MA-induced locomotor activation in mice. Further, passive immunization with ABs decreases cardiovascular responses to MA and blocks MA-induced reinstatement of extinguished lever press responding in rats.

The use of TT as the protein carrier with the SMA hapten is important because, when SMA is conjugated to diphtheria toxoid, it produces a weaker immune response [19]. However, the strong immune response seen in the present study with TT appears to be due to the Toll-like receptor-5 (TLR-5) adjuvant, entolimod [43] combined with alum. The anti-MA IgG AB levels reach a maximum level with the 1 µg entolimod dose plus alum and are over 35% higher (964 μg/mL vs. 1519 μg/mL) compared to the vaccine with alum alone or compared to the lower entolimod dose of 0.3 µg (940 μg/kg; see Figure 1). Furthermore, entolimod doses higher than 1 µg do not lead to greater peak anti-MA AB levels nor to an enhanced ability to sustain anti-MA AB levels over weeks. Finally, the TT-SMA plus entolimod vaccine does not reveal any obvious toxicity other than a local injection site skin induration in a few mice that disappeared within a relatively short time. In fact, there are no untoward cardiovascular effects seen in rats to the combination of passively administered anti-MA ABs and MA (see Figure 3). Passive administration of anti-MA ABs did not alter urine production or water intake or affect urine MA detection (data not shown). All animals received the prescribed dosage without complications. 

The functional significance of the anti-MA AB is clearly shown in mice in the present study by its ability to reduce MA-induced locomotor activity. Not only is total locomotor activity across the 90-min session significantly lower in vaccinated mice, there is a smaller peak locomotor response that begins to return to baseline more quickly than the response seen in the non-vaccinated mice after MA administration (see Figure 2). Furthermore, lower locomotor activity is significantly correlated with higher anti-MA AB levels and explains 62% of the variance. Assessing locomotor responses to MA is a sensitive, convenient, and oft-used method to test whether the vaccine has functional effects. We [14,21] and others [12,44,45,46] employed such tests in rodents to demonstrate the functional efficacy of various anti-MA vaccines. We employed locomotor tests of MA administration in mice in our previous research and find that the dose tested in the present study (2 mg/kg) reliably produces increased activity levels that typically peak between 10- and 20 min post-administration and remain above baseline levels for up to 90 min (refs. [14,21]). In fact, lower MA doses (<0.1 mg/kg) decrease activity levels and higher doses (>3.0 mg/kg) lead to large increases in activity initially (10 min) that drop below baseline levels at 20 to 30 min post administration [47]. Not all MA vaccines result in reductions in the functional activity of MA. For example, an anti-MA vaccine linked to KLH does not alter the locomotor effects of a moderately high MA dose (3 mg/kg) of MA [8], perhaps due to the interactions of age, strain, species, test apparatus, and MA dose [48].

MA stimulates the sympathetic nervous system leading to increases in blood pressure and heart rate acutely and chronic use is linked to cardiovascular pathology [49]. An anti-MA vaccine that could protect against large cardiovascular responses to MA would be advantageous. Thus, we tested whether passive immunization with anti-MA ABs could reduce some cardiovascular effects of MA in rats. Results show some support for this notion. MA (2 mg/kg; IV) increased mean arterial pressure (MAP) in both male and female rats with the change in MAP response decreasing across subsequent dosings over time (see Figure 3). The MA-induced increase in MAP was significantly attenuated by administration of the anti-MA ABs in rats of both sexes. In contrast, the MA-induced increase in HR, seen in rats of both sexes, was not altered by passive administration of anti-MA ABs. Although we did not anticipate that the anti-MA ABs would enhance the cardiovascular effects of MA, that we saw a reduction in the MAP response in rats of both sexes and no effect on HR response suggests that this combination is not toxic. Further, it shows the likelihood that passive administration of anti-MA ABs would be well-tolerated in humans.

The administration of serum anti-MA ABs derived from rats vaccinated with TT- SMA plus entolimod significantly reduces the ability of an MA injection to reinstate extinguished lever press responding, a major test of the vaccine’s potential therapeutic efficacy. Moreover, three out of four rats emitted no lever presses in response to the MA injection after passive AB administration in sharp contrast to the high level of responding seen prior to AB administration. The reinstatement test in operant self-administration is often used to model relapse to use after abstinence in humans [38,39,40]. Because these tests are conducted after much drug exposure and in animals that are well-trained to perform an operant to obtain the drug, it provides a relevant test of the efficacy of the anti-MA vaccine. Previously, we showed that MA-induced reinstatement of place conditioning is reduced in mice after active vaccination with TT-SMA and, in fact, MA produced a conditioned place aversion in some mice [17]. While it is not possible to assess an aversive response to MA in reinstatement of operant responding, the results of the present study in rats are consistent with our prior work in mice. Passive administration of anti-cocaine ABs was used in prior studies that report similar results [50,51]. Active immunization with another anti-MA vaccine also blocks MA-induced reinstatement but only in male rats [23], in contrast to the present study that shows effects in rats of both sexes. This vaccine and another also lead to reductions in acquisition or maintenance of MA self-administration in rats [18,52] but in another case, an anti-MA vaccine enhanced MA self-administration [9]. The authors of the latter study suggest that the increased MA self-administration in vaccinated rats reflects pharmacokinetic antagonism akin to results seen with pharmacodynamic antagonism [53,54].

It is possible that vaccination with TT-SMA plus entolimod induced Th17 cells that appear to form long-term post-vaccination memory cells for continued antibody responses to vaccinations [55]. Furthermore, concentrations of Th17 cell populations pre-vaccination could be useful for the diagnosis, staging, and monitoring of the therapeutic efficacy with various vaccination strategies [56]. Murdaca et al. (2019) [57] also suggested that the IL-31/IL-33 axis represents a potential pathway for sustaining the antibody response to vaccinations, just as this IL-33/ST2 and Th2/IL-31 immune response pathway is involved in developing allergic inflammation. In the future, these cytokines and Th17 cells might be induced directly or used as adjuvants to increase and sustain antibody responses to vaccination.

## 5. Conclusions

In conclusion, the combination of TT-SMA and alum with entolimod adjuvant produced a robust anti-MA AB response that attenuates MA-induced locomotor activity in mice and reinstatement of self-administration behavior in rats without any untoward cardiovascular effects. Adding entolimod improves the efficacy of the TT-SMA vaccine, compared to alum alone, making this combination vaccine an excellent candidate for development and testing in human clinical trials for MUD.

## Figures and Tables

**Figure 1 vaccines-10-01508-f001:**
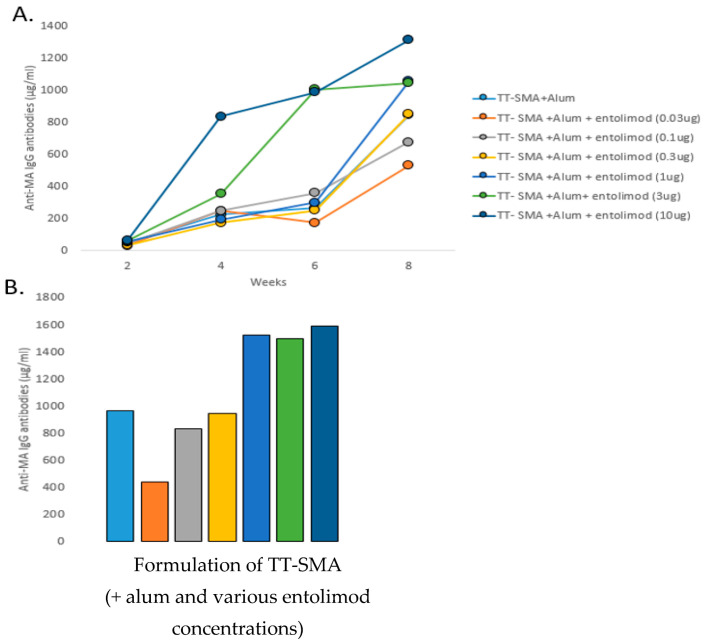
Immunopotentiating effects of entolimod adjuvant. (**A**) IgG anti-MA antibody levels in mice vaccinated with TT-SMA (32 ug) in the presence and absence of alum or entolimod in serum collected at 2, 4, 6, and 8 weeks post initial vaccination. Data are presented as levels of antibody in pooled samples from each group (n = 5/group). (**B**) Week 10 anti-MA antibody levels across all groups.

**Figure 2 vaccines-10-01508-f002:**
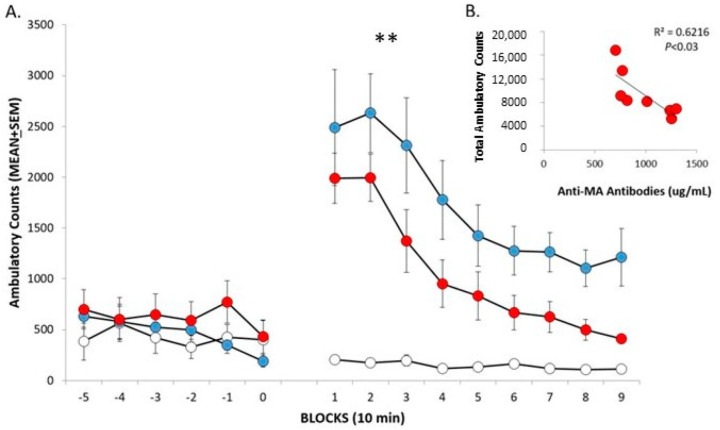
MA-induced locomotor activation. (**A**) Mice were first habituated to the apparatus (-5-0) then unvaccinated mice were administered saline (white circles, 0 mg/kg, n = 5) or MA (blue circles, 2 mg/kg, n = 10) and vaccinated mice received MA (red circles, 2 mg/kg, n = 8). MA-induced locomotor activation was significantly attenuated in vaccinated mice compared to unvaccinated mice. ** main effect for vaccine, p < 0.0001. (**B**) Anti-MA antibody levels significantly correlated with total ambulatory counts.

**Figure 3 vaccines-10-01508-f003:**
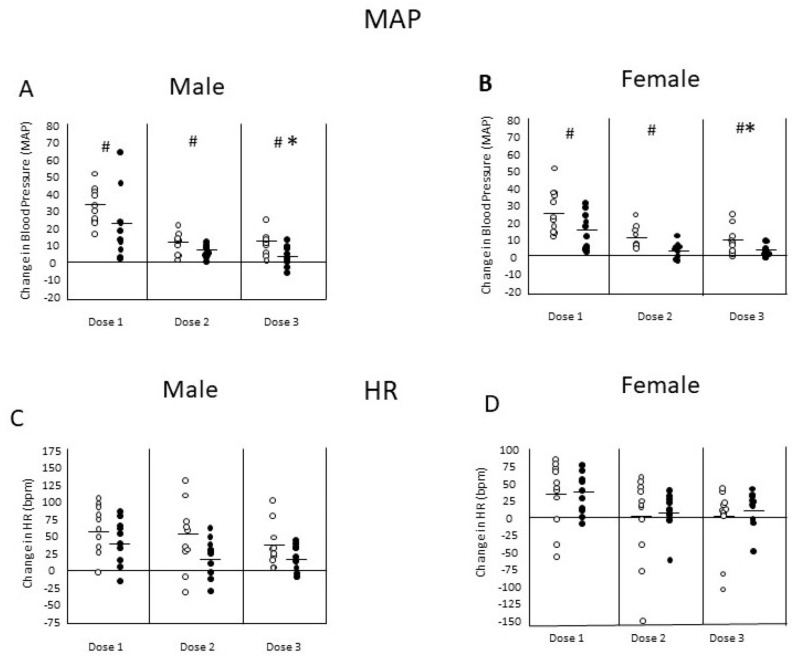
Changes in mean arterial pressure (MAP) and heart rate (HR) elicited by 3 consecutive doses of MA (2 mg/kg, i.v.) in male (n = 10) and female (n = 11) Sprague-Dawley rats, prior to and after passive immunization with anti-MA antibodies. (**A**,**B**) changes in MAP elicited by MA in male and female rats, respectively. Pressor responses were significantly (* = *p* < 0.05) reduced between the first and third MA dosings in male and female rats. After immunization, the pressor responses to MA were significantly (# = *p* < 0.05) smaller than during control (treatment effect). (**C**,**D**) changes in HR elicited by MA in male and female rats, respectively. There was a significant effect of dose on HR, but no significant differences in the HR responses before and after immunization (treatment effect). There was no significant interaction between dose and treatment.

**Figure 4 vaccines-10-01508-f004:**
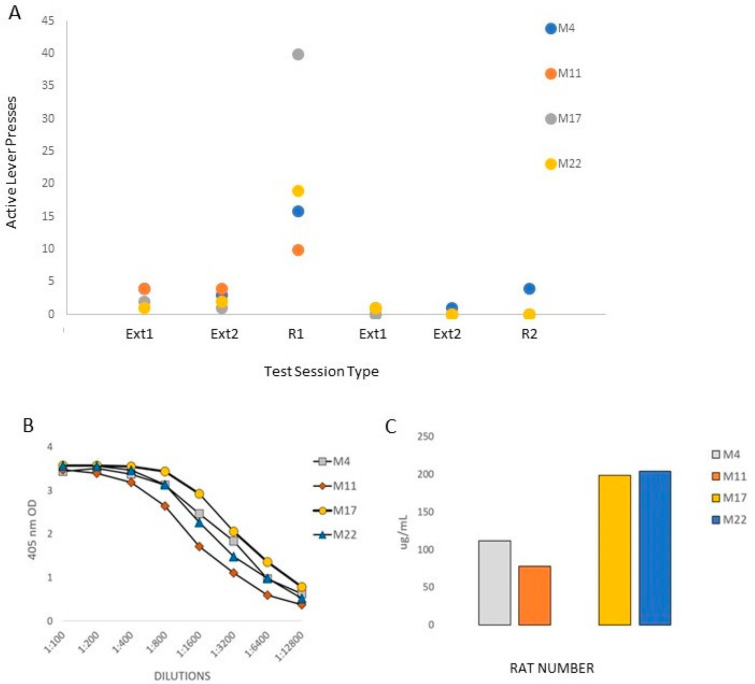
Passive immunization with anti-MA antibodies blocks drug-induced reinstatement of MA-seeking behavior. Data presented are from two female (M4 and M11) and two male (M17 and M22) rats that were tested at baseline then after being passively immunized with antibodies generated from the TT-SMA vaccine with entolimod. (**A**) After rats underwent extinction (EX1-EX2), rats received an IP injection of MA (2 mg/kg; R1) that produced a significant increase in lever pressing (i.e., drug-seeking). Following reinstatement tests, extinction training resumed and rats were passively immunized with anti-MA antibodies. After passive immunization, rats received another dose of MA (R2). Passive immunization with anti-MA antibodies significantly blocked drug-seeking behavior and three of four rats that showed no lever pressing. (**B**) Serial dilutions of serum from each rat and (**C**) the total anti-MA antibody levels are presented.

## Data Availability

The data presented in this study are available on request from the corresponding author.

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
