# Peer review of "Active and Passive Immunization with an Anti-Methamphetamine Vaccine Attenuates the Behavioral and Cardiovascular Effects of Methamphetamine"

_vaccines, 2022, doi:10.3390/vaccines10091508_

Round 1
Reviewer 1 Report
In this manuscript, the authors claimed that they combined TT-SMA and alum as adjuvants to induce strong anti-MA antibody which can attenuate MA-induced locomotor activity without affecting cardiovascular function. They effort to explore new vaccine in human clinical trials for MUD. But, the experimental design of this paper still has some issues. The detailed comments are as follows.
Specific comments:
1. Evaluation of antibodies in different aspects, i.e. the affinity and titer of antibody can provide more information to support the given statements of anti-MA response ability.
2. In the MA-induced locomotor activation test, the author selected the specific movement distance and time to prove the activity level of mice. What’s the detailed consideration about this selection? Whether other time duration or distance could be also used for further identification to make the results more reliable? Moreover, is it possible to add more index in this part to prove the MA’s effect on locomotor?
3. For experiments described about the anti-MA antibody purified from rats there is no information on its characterization. Please include this information before MA Cardiovascular Effects detection after vaccine treatment.
4. Were all antibody purified from the rats performed endotoxin removal and corresponding safety evaluation? Please clarify.
5. In figure 3, the authors need to show individual data points in the graph to make the data look more reliable.
6. On the Figure 4C, x-axis there are no values. Please revise, expand the methods section to include all the details of data in Figure 4.
Author Response
We thank the two Reviewers for their helpful comments on the manuscript. We have revised the paper in accordance with their suggestions as detailed below.
Reviewer #1
- Evaluation of antibodies in different aspects, i.e. the affinity and titer of antibody can provide more information to support the given statements of anti-MA response ability.
We added text to the first paragraph of the Discussion that describes our prior work in which we characterized the affinity of TT-SMA and examined antibody titers (from ref # 21). Please see yellow highlighted text.
- In the MA-induced locomotor activation test, the author selected the specific movement distance and time to prove the activity level of mice. What’s the detailed consideration about this selection? Whether other time duration or distance could be also used for further identification to make the results more reliable? Moreover, is it possible to add more index in this part to prove the MA’s effect on locomotor?
We added text to the third paragraph of the Discussion that describes the reliability of the locomotor measure. Please see yellow highlighted text. We have been conducting studies using locomotor activity measures for over 30 years with rats and mice, assessing various drugs, and utilizing different apparatuses. Essentially, horizontal movement or distance traveled are correlated and thus, from our prior experience, using more than one measure does not add any additional information. Other measures, such as center time or rearing, do not necessarily correlate with movement or distance traveled but these measures were not relevant for this study.
- For experiments described about the anti-MA antibody purified from rats there is no information on its characterization. Please include this information before MA Cardiovascular Effects detection after vaccine treatment.
We added text to the end of the paragraph in the Methods section on passive administration to state that the anti-MA antibody levels were quantified by ELISA (2.5).
- Were all antibody purified from the rats performed endotoxin removal and corresponding safety evaluation? Please clarify.
This is a valid point. We did not perform endotoxin removal or evaluate safety studies however, we did not see any evidence that passive immunization with sera containing anti-mA antibodies had adverse effects based on the health of the rats. We also obtained weights which either did not change or increased over the duration of the experiments.
- In figure 3, the authors need to show individual data points in the graph to make the data look more reliable.
Figure 3 has been re-drawn to include individual data points.
- On the Figure 4C, x-axis there are no values. Please revise, expand the methods section to include all the details of data in Figure 4.
The x-axis in Figure 4C has been added. We also added text to the Methods section (2.9) that provides more procedural detail. Please see yellow highlighted text.
Reviewer 2 Report
The paper is interesting and well written. I suggest to discuss the role of Th17 cells and IL-31/IL-33 axis after vaccinations (see and add as references papers by Muirdaca et al concerning these topics)
Author Response
We thank the two Reviewers for their helpful comments on the manuscript. We have revised the paper in accordance with their suggestions as detailed below. All major changes are highlighted in yellow.
Reviewer #1
- Evaluation of antibodies in different aspects, i.e. the affinity and titer of antibody can provide more information to support the given statements of anti-MA response ability.
We added text to the first paragraph of the Discussion that describes our prior work in which we characterized the affinity of TT-SMA and examined antibody titers (from ref # 21). Please see yellow highlighted text.
- In the MA-induced locomotor activation test, the author selected the specific movement distance and time to prove the activity level of mice. What’s the detailed consideration about this selection? Whether other time duration or distance could be also used for further identification to make the results more reliable? Moreover, is it possible to add more index in this part to prove the MA’s effect on locomotor?
We added text to the third paragraph of the Discussion that describes the reliability of the locomotor measure. Please see yellow highlighted text. We have been conducting studies using locomotor activity measures for over 30 years with rats and mice, assessing various drugs, and utilizing different apparatuses. Essentially, horizontal movement or distance traveled are correlated and thus, from our prior experience, using more than one measure does not add any additional information. Other measures, such as center time or rearing, do not necessarily correlate with movement or distance traveled but these measures were not relevant for this study.
- For experiments described about the anti-MA antibody purified from rats there is no information on its characterization. Please include this information before MA Cardiovascular Effects detection after vaccine treatment.
We added text to the end of the paragraph in the Methods section on passive administration to state that the anti-MA antibody levels were quantified by ELISA (2.5).
- Were all antibody purified from the rats performed endotoxin removal and corresponding safety evaluation? Please clarify.
This is a valid point. We did not perform endotoxin removal or evaluate safety studies however, we did not see any evidence that passive immunization with sera containing anti-mA antibodies had adverse effects based on the health of the rats. We also obtained weights which either did not change or increased over the duration of the experiments.
- In figure 3, the authors need to show individual data points in the graph to make the data look more reliable.
Figure 3 has been re-drawn to include individual data points.
- On the Figure 4C, x-axis there are no values. Please revise, expand the methods section to include all the details of data in Figure 4.
The x-axis in Figure 4C has been added. We also added text to the Methods section (2.9) that provides more procedural detail. Please see yellow highlighted text.
Reviewer #2
The paper is interesting and well written. I suggest to discuss the role of Th17 cells and IL-31/IL-33 axis after vaccinations (see and add as references papers by Muirdaca et al concerning these topics)
An additional paragraph has been added at the end of the Discussion to discuss this interesting point. Please see yellow highlighted text.
Round 2
Reviewer 1 Report
All my questions are well responsed.